# Long-Term Outcomes after Incomplete Resection of Intramedullary Grade II Ependymomas: Is Adjuvant Radiotherapy Justified?

**DOI:** 10.3390/cancers15143674

**Published:** 2023-07-19

**Authors:** Elly Chaskis, Mohamed Bouchaala, Philippe David, Fabrice Parker, Nozar Aghakhani, Steven Knafo

**Affiliations:** 1Department of Neurosurgery, Bicêtre Hospital, AP-HP, 94270 Le Kremlin-Bicêtre, France; elly.chaskis@aphp.fr (E.C.);; 2Faculty of Medicine, University Paris-Saclay, 94270 Le Kremlin-Bicêtre, France

**Keywords:** spinal cord, ependymoma, surgery, radiotherapy

## Abstract

**Simple Summary:**

The long-term outcomes and indications for radiotherapy after incomplete removal of grade II intramedullary ependymomas remain unclear. In this retrospective single-center study including 46 patients, no adjuvant radiotherapy was performed, even after subtotal resection (n = 21/46). Over a median follow-up of 6.5 years, radiological progression of a tumoral remnant was noted in seven patients, including only two who were symptomatic and four who required further treatment. Altogether, progression-free survival was 90% at 5 years and 76.8% at 10 years. Based on our experience, we suggest a management algorithm for patients with grade II intramedullary ependymomas where no adjuvant radiotherapy is proposed on a systematic basis, even after subtotal resection. Whenever the tumor can be safely resected, gross total resection should be the aim; otherwise, subtotal removal without adjuvant radiotherapy seems a reasonable option to maximize patient safety with a low risk of clinical progression.

**Abstract:**

Ependymomas are the most common intramedullary tumors in adults. While gross total resection is the aim of surgery, tumor infiltration might limit resection. In cases of subtotal removal, the necessary adjuvant management remains unclear. The aim of our study was to assess the need for adjuvant radiotherapy after an incomplete resection of grade II intramedullary ependymomas (IME-II). We retrospectively reviewed all cases of IME-II operated upon at a single tertiary neurosurgical center from 2009 to 2018. Patients with anaplastic or myxopapillary ependymomas, and patients with a follow-up of less than three years, were excluded. We included 46 patients: 19 (41.3%) had a gross total resection; 21 (45.7%) had a subtotal resection; and 6 (13%) had a partial resection. None of the patients underwent adjuvant radiotherapy. Over a median follow-up of 79 months (range = 36–186), seven patients presented a radiological tumor progression with a mean delay of 50.9 months (range = 18–85), of which two were symptomatic (4.3%). Progression-free survival (PFS) was 90.1% at 5 years and 76.8% at 10 years. The extent of the resection was the only significant risk factor for secondary tumor progression (*p* = 0.012). Four of the seven patients with recurring IME-II were treated: three patients had a second surgery, leading to two GTR and one STR, followed by radiotherapy in one case, and one patient underwent radiotherapy alone. In this study, the rate of symptomatic progression and retreatment after incomplete resection of IME-II without adjuvant radiotherapy was low, suggesting a conservative approach in such cases.

## 1. Introduction

Ependymomas are the most frequently occurring intramedullary tumors in adults, accounting for 60 to 70% of cases [1,2,3]. Intramedullary ependymomas (IME) arise from ependymal cells in the central canal of the spinal cord and are more frequently located in the cervical region [4,5]. Most IMEs are benign and slowly progressing tumors, classified as grade II in the WHO classification, with anaplastic (grade III) tumors representing approximately 5% of IMEs [3,6].

For patients with symptomatic tumors, surgery is the treatment of reference. Its aim is maximal resection but also preservation of neurological function [4,5]. While gross total resection (GTR) is reported in 79.2 to 91.5% of the patients [1,2,4,5,6,7], subtotal resection (i.e., resection of more than 90% of tumor volume) might be an acceptable strategy to maximize patient safety whenever the tumor infiltrates the spinal cord [8,9].

The indication for adjuvant radiotherapy (RT) after surgery for IME-II remains unclear. Several studies have recommended systematic postoperative RT after incomplete resection of an IME-II [3,6,10,11], while others did not demonstrate increased progression-free survival (PFS) after irradiation [2,12,13]. Guidelines from the European Association of Neuro-Oncology (EANO) recommend postoperative RT (45–54 Gy) after incomplete resection of an IME-II but not after gross total resection [14]. However, this recommendation is based on a pooled analysis of the literature [10] and ignores a retrospective cohort published simultaneously that found no increase in progression-free survival due to adjuvant RT after an incomplete resection [12].

Therefore, the aim of our study was to evaluate the risk of progression after incomplete removal of an IME-II without adjuvant RT.

## 2. Materials and Methods

Patients operated on for an IME-II between 2005 and 2018 in the Department of Neurosurgery of Bicêtre Hospital (University Paris-Saclay) were screened. We retrospectively collected all preoperative, operative, and postoperative clinical and imaging data from medical records, surgical reports, and follow-up MRIs. We excluded cases without at least yearly postoperative Magnetic Resonance (MR) imaging follow-up for 3 years after surgery. Patients with grade III ependymomas were excluded since IME-III prognostics and management are different from IME-II’s [3,15]. Neuropathological diagnosis was based on the 2016 WHO classification; no molecular profiling was routinely performed.

Extension of resection was classified according to both the surgical report and an early (within 3 months postoperative) MRI. Surgery was classified as gross total resection (GTR) when complete resection was reported by the surgeon and no residual tumor was visible on an MRI; subtotal resection (STR) when a small residual tumor was leftover according to the surgical report and measured as being less than 10% of the initial tumor volume on an MRI; and partial resection (PR) when it was larger than 10% of the initial tumor volume on an MRI.

Clinical status was evaluated according to the modified McCormick scale [16]. A radiological tumoral growth on an MRI during follow-up was classified as tumor progression, even in the absence of new neurological symptoms related to this tumoral progression.

Statistical analysis was performed using IBM^®^ SPSS^®^ Statistics v22, using the Kaplan–Meier estimator for survival analysis, the log-rank test for identification of risk factors, and Cox regression analysis for multivariate analysis. A *p*-value of 0.05 or less was considered statistically significant.

## 3. Results

### 3.1. Patients

We screened 75 patients operated on for an intramedullary ependymoma in our department between January 2005 and December 2018. Five were grade III ependymomas; twenty-three patients were lost to follow-up; and one patient died from ovarian adenocarcinoma 25 months after the IME surgery. Therefore, we included 46 out of 68 eligible patients with IME-II.

The mean age was 45.5 years old (range = 24–69); there were 22 women (47.8%) and 24 men (52.2%). The onset of symptoms was progressive for all patients, with a mean delay of 30.3 months before surgery (range = 1–204). Sensory deficits were the most frequent presenting symptom, reported by 93.5% of the patients, followed by pain (82.6% of patients), motor weakness (41.3%), and sphincter disturbances (30.4%). Most patients (95.7%) were able to walk without aid before surgery (McCormick grade I and II), with only four patients requiring external assistance (McCormick grade III and IV). Radiologically, tumors extended over three vertebral levels on average (range 1–6); all were enhanced after gadolinium injection on T1-weighted sequences; 86.7% were associated with an intramedullary cyst and 58.6% to a hemorrhage, usually visible as a peritumoral T2 hyposignal; there was no distant spinal or cerebral localization (Table 1).

### 3.2. Surgery

Most surgical procedures (37/46) were performed without intraoperative electrophysiological monitoring. According to surgical reports, a clear dissection plane between the tumor and the spinal cord was conserved throughout the surgery in less than half of the patients (21/43). Consequently, resection was reported by the surgeon as incomplete in 27/46 patients (58.7%), usually in the form of a thin tumoral layer left over the spinal cord where the dissection plane was lost (STR = 21/46). All tumors included in this study were classified as grade II ependymomas according to the WHO classification, with Ki-67 levels < 5% in 90.9% of the cases (Table 2).

Immediate postoperative examination demonstrated clinical worsening in 41.3% of patients: while most patients remained ambulatory (McCormick grades I and II, from 95.7% preoperatively to 78.2% postoperatively), almost one patient in five presented a more severe neurological deficit requiring aid or assistance for ambulation (McCormick grades III and IV, from 4.4% to 21.7%). 

Four patients developed postoperative complications: one patient presented a pseudo-meningocele requiring surgical revision; one patient developed a pulmonary embolism; one patient had pulmonary atelectasis; and one patient presented a urinary tract infection. 

Early postoperative MR imaging was performed after a median time of 4.5 months (range = 0–53). A residual tumor was visible on this first postoperative MRI in 22 cases, even though there were 27 incomplete resections, according to surgical reports, that were therefore considered as such. 

Even in cases of incomplete resections, no systematic radiotherapy was performed, and all patients were initially followed up conservatively with yearly clinical examination and an MRI. 

### 3.3. Follow-Up

Patients were followed up for a median duration of 79 months (mean = 86.7, range = 36–186). At the last follow-up evaluation, most patients (84.8%) were independent (McCormick grades I and II). Compared to their preoperative status, the individual McCormick grade was unchanged in 29 patients (63%), improved in six patients (13%), and worsened in 11 patients (24%), including two grade IV patients who both were grade II preoperatively. There was no correlation between the extent of resection and the risk of postoperative neurological impairment.

Radiological tumor progression was observed in seven patients (15.2%) with a mean delay of 50.9 months after surgery (range = 18–85), including six cases of subtotal resections and one case of partial resection (Figure 1). None of the patients with gross total resection recurred. Interestingly, two of the recurring patients were considered as subtotal resections by the surgeon, even though no residual tumor was visible on their early postoperative MRIs. Only two patients developed sensory symptoms due to tumor progression, while the five other cases remained asymptomatic. Progression-free survival (PFS) rates for IME-II were therefore 93.4% at 3 years; 90.1% at 5 years, and 76.8% at 10 years (Table 2 and Figure 1).

Of the seven patients with radiological tumor progressions, four were treated, including the two symptomatic cases. Three patients were reoperated upon: two had a second subtotal resection, while complete resection was achieved in the remaining case. Surgery was followed by radiotherapy in one patient, and one patient received radiotherapy without surgery. Among these four patients who were re-treated, three remained stable with a follow-up period of, respectively, 3, 3, and 8 years (Figure 2). The patient treated by surgery and adjuvant radiotherapy developed another tumor progression 3 years after re-treatment. Three patients with small and asymptomatic radiological progressions were managed conservatively, without further progression to date (Table 3).

Predicting factors for PFS were the extent of resection at first surgery (log-rank, chi-2 = 8.9, *p* = 0.012) and female gender (log-rank, chi-2 = 4.5, *p* = 0.034). Neither preoperative McCormick scores, nor the size of the initial tumor, nor Ki-67 levels were predictive of tumor progression. 

## 4. Discussion

In this retrospective single-center study including 46 patients who had surgery for an IME-II, we aimed to interrogate the progression rate after incomplete removal without adjuvant radiation therapy. 

Indeed, although IME-IIs are usually well circumscribed and amenable to gross total resection, they are not encapsulated lesions and may sometimes be infiltrative [4,8,17]. In such cases, a subtotal resection can be performed to maximize safety and will, in most cases, consist in leaving a thin tumoral layer wherever the cleavage plan is unclear. This small tumoral remnant might not even be visible on the early postoperative MRI (as was the case in five of the patients included in our series), explaining why GTR rates based solely on postoperative MRIs are probably overestimated. 

In our experience, and based on both surgical reports and postoperative MRIs, GTR was achieved in 41.3% (19/46) of patients, while STR was performed in 45.7% of cases (21/46). This rate of GTR might seem significantly lower than what has been reported in other large monocentric series (86.3% of GTR in Klekamp [4], 83% in Bostrom et al. [7], for instance) in which the extent of resection was assessed based on postoperative MRIs. However, it is close to what has been reported in studies evaluating the extent of resection based on both MRIs and surgical reports (e.g., 54% of GTR in Savoor et al. [6]). Since the STR of IME-IIs usually consist in a thin tumoral remnant that might not be visible on early MRIs, we recommend using surgeons’ self-assessments as a more reliable estimate of the extent of resection. 

Partial resections (<90%) of IME-IIs are rare (13% in our series) and are often the result of an unusual presentation of the tumor limiting surgical resection. Such cases are therefore best treated with adjuvant radiotherapy. In this case series, no adjuvant radiotherapy was indicated after STR. Over a median follow-up of 6.5 years with yearly MRIs, radiological progression of a tumoral remnant was noted in only seven patients, including two who were symptomatic and four who required further treatment. Altogether, progression-free survival was 90% at 5 years and 76.8% at 10 years, in line with previous case series [2,4,5,7].

Notably, the PFS observed in our study, in which no adjuvant radiotherapy was proposed after incomplete removal, is similar or even superior to studies in which radiotherapy was performed after STR (10 years’ PFS for STR + RT = 50% [11] to 77.1% [6]). One explanation is that these studies, as others [3,5], mixed ependymoma subtypes (e.g., myxopapillary and anaplastic) with very different prognoses. Although RT is commonly indicated for grade III ependymomas, even after complete resection, whether it is beneficial in grade II tumors remains controversial [13].

Of course, our study is a single-center retrospective series that does not allow us to draw an affirmative conclusion regarding whether adjuvant radiotherapy is effective or not to prevent recurrence after an incomplete resection of an IME-II. To do so, a randomized multicenter trial would be necessary. However, given the rare occurrence and slow growing rate of these tumors, a more feasible approach would be to perform a propensity-matched cohort study. However, even such methods might not provide conclusive evidence, as was recently the case regarding the effect of RT after STR in intracranial ependymomas [18].

Moreover, RT toxicity should also be considered regarding the best timing for irradiation. Indeed, the risk of radiation myelopathy after stereotactic RT, estimated between 1% for 17 Gy delivered in two fractions to 5% for 25.3 Gy delivered in five fractions, increases in cases of reirradiation [19]. In our experience, most patients with STR of an IME-II will not progress over a relatively long follow-up. And if their tumor eventually progresses, it will be possible to proceed to RT, with or without a second surgery, which would most likely not be the case had the patient already been irradiated after the initial surgery. 

Recurrence after surgery for IME-II has been reported in 2.7 to 8.1% of patients, but its risk factors remain unclear [2,3,7,20]. In recurring cases, reoperation, radiation therapy, or chemotherapy are available options [3,11,12]. In our study, the main predicting factor for progression was the extent of resection, as was also reported by others [2,7]. Notably, there was no progression among the patients for which gross total resection could be achieved. This finding confirms that whenever complete resection can be performed safely, it remains the best surgical strategy for IME-II regarding oncological outcome. 

Although a significant number of patients demonstrated some neurological impairment in the early postoperative period as expected [1,17], severe deterioration remained rare (13% of new McCormick grade III/IV). The absence of correlation between the extent of resection and postoperative neurological problems reflects our surgical strategy. Whenever the cleavage plane between the tumor and the spinal cord is preserved, GTR should be performed; otherwise, STR with leaving a thin layer of tumor overlying the invaded spinal cord tissue seems a reasonable option.

Recently, molecular studies of ependymomas have led to reappraisal of these tumors in the WHO 2021 classification [21]. In particular, *MYCN* amplification now defines a distinctive subtype of aggressive spinal ependymoma with early leptomeningeal dissemination and anaplastic morphology [22]. Moreover, spinal myxopapillary ependymomas are now classified as WHO grade II tumors since their prognosis is more consistent with grade II behavior [21]. Lastly, methylation-based classification also allows reclassification of spinal ependymomas to molecular subgroups, although no significant clinical and prognostic features have yet emerged [23]. No doubt, future molecular studies will advance our understanding of spinal ependymomas and might refine our therapeutic strategy, notably regarding the indication of adjuvant radiotherapy.

## 5. Conclusions

Based on our experience, we propose the following algorithm for managing patients with IME-II (Figure 3). Surgery is indicated for evolutive and/or symptomatic tumors, and GTR should always be the aim of this initial surgery [14]. However, if the cleavage plane between the tumor and the spinal cord does not allow for safe complete removal, STR can be performed by leaving a thin layer of tumor overlying the spinal cord. Whether GTR or STR was achieved, we propose, based on our experience, that no adjuvant radiotherapy be performed on a systematic basis and that patients are followed up yearly with an MRI. If a radiological recurrence is observed, a second surgery is proposed, if deemed feasible by the surgeon and especially if the patient is symptomatic. Adjuvant radiotherapy can be performed after this second surgery, particularly if GTR could not be achieved.

This strategy departs from many studies [3,6,10,11] and the EANO guidelines [14] recommending systematic adjuvant radiotherapy after incomplete resection of a spinal IME-II. Even though the observational design of the present study does not allow us to draw any conclusion regarding the efficacy of adjuvant RT for IME-II, it nonetheless underscores the low radiological and even lower clinical progression rate of these tumors. Therefore, we believe that preserving neurological function and minimizing iatrogenicity should be driving the management of these patients, who are often young and professionally active. Whenever the tumor can be safely resected, GTR should be the aim; otherwise, STR without adjuvant radiotherapy seems a reasonable option to maximize patient safety.

## Figures and Tables

**Figure 1 cancers-15-03674-f001:**
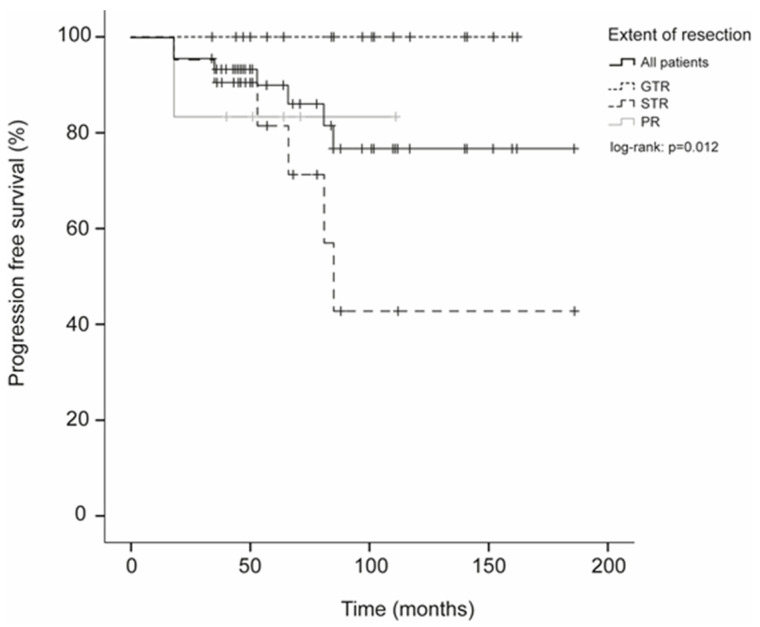
Survival analysis for grade II intramedullary ependymomas.

**Figure 2 cancers-15-03674-f002:**
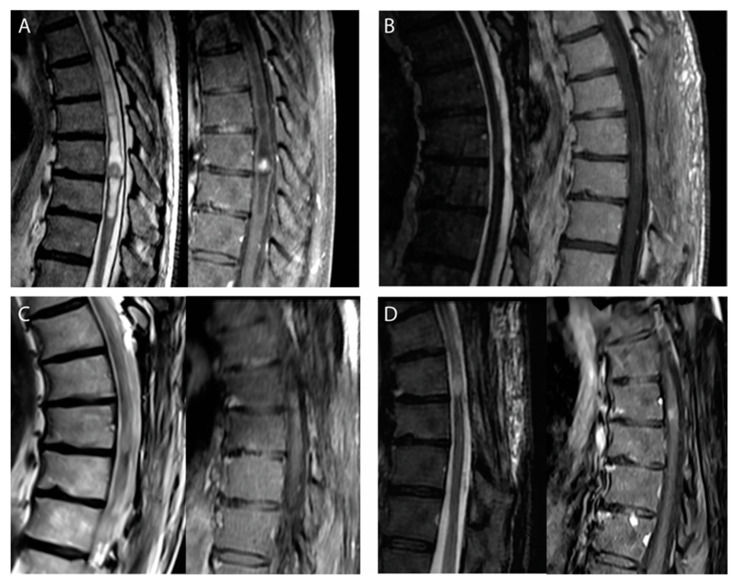
Illustrative case of a recurring IME-II managed conservatively. (**A**) Preoperative MRI (sagittal T2, **left**; T1 with gadolinium, **right**) demonstrating a T8–T9 IME-II with peritumoral cysts in a 59-y.o. patient. (**B**) Early postoperative MRI at 3 months showing no residual tumor although the surgeon considered the resection as subtotal due to the loss of the cleavage plane anteriorly. (**C**) Follow-up MRI at 48 months after surgery showing a discrete radiological progression with a millimetric enhancement. (**D**) The last follow-up MRI performed 67 months after surgery confirming this slowly progressing progression, which was managed conservatively in the absence of any new symptom.

**Figure 3 cancers-15-03674-f003:**
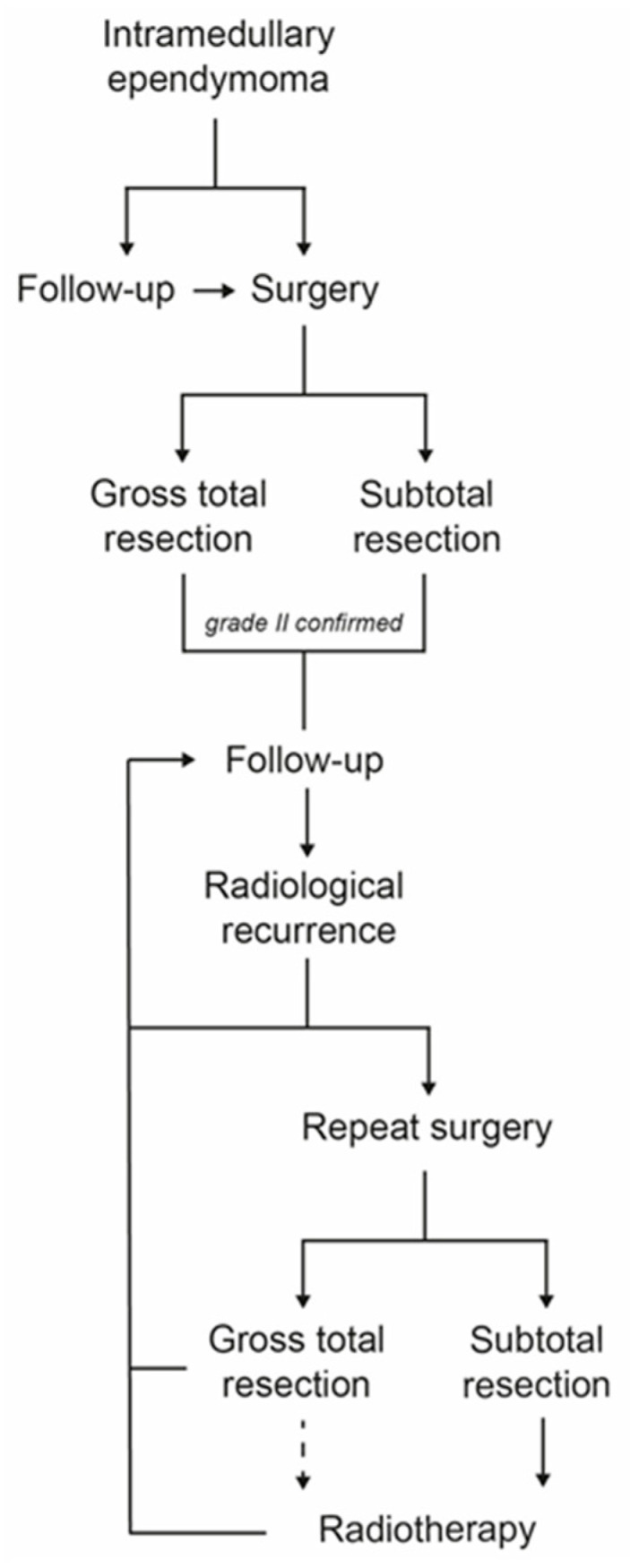
Suggested management algorithm for grade II intramedullary ependymomas.

**Table 1 cancers-15-03674-t001:** Baseline clinical and radiological features.

Clinical Features		
Mean age (years, [min–max])		45.5 [24–69]
Sex ratio (M/F)		24/22 (52.2%)
Mean delay between symptom onset and surgery (months, [min–max])		30.3 [1–204]
Preoperative McCormick grade	I	25 (54.4%)
	II	19 (41.3%)
	III	2 (4.4%)
	IV	0
Preoperative symptoms	pain	38 (82.6%)
	motor	19 (41.3%)
	sensory	43 (93.5%)
	sphincter	14 (30.4%)
**Radiological Features**		
Vertebral levels involved	mean [min–max]	3 [1–6]
	cervical	20 (43.5%)
	cervicodorsal	8 (17.4%)
	dorsal	18 (39.1%)
T1-weighted sequence signal (%)	hypo	54.8%
	iso	25.0%
	hyper	16.1%
T2-weighted sequence signal (%)	hypo	17.1%
	iso	25.7%
	hyper	57.1%
Gadolinium enhancement		100%
Peritumoral cyst		86.7%
Peritumoral hemorrhage		58.6%

**Table 2 cancers-15-03674-t002:** Surgical, postoperative, and long-term outcomes.

Surgical Outcome			
Clear cleavage plane throughout surgery		21/43 (48.8%)	
Residue according to surgeon		27 (58.7%)	
Residue visible on first postop MRI		22 (47.8%)	
Extent of resection	gross total resection	19 (41.3%)	
	subtotal resection	21 (45.7%)	
	partial resection	6 (13%)	
WHO grade	II	46 (100%)	
	III *	0	
Ki-67 levels	<5%	40 (90.9%)	
	≥5%	4 (9.1%)	
**Postoperative Outcome**			
Early postoperative McCormick grade	I	14 (30.4%)	
	II	22 (47.8%)	
	III	7 (15.2%)	
	IV	3 (6.5%)	
Complications	medical (PE, UTI)	2/46 (4.3%)	
	reoperation < M3	2/46 (4.3%)	
**Long-Term Outcome**			
Follow-up	Mean (in months)	86.7	
	Median [min–max]	79 [36–186]	
McCormick grade at last follow-up	I	32 (69.6%)	
	II	7 (15.2%)	
	III	5 (10.9%)	
	IV	2 (4.3%)	
Tumor progression	Mean delay [min–max]	50.9 [18–85]	
	Radiological	7 (15.2%)	
	Symptomatic	2 (4.4%)	
	Progression free survival (PFS)	3 years	93.4%
		5 years	90.1%
		10 years	76.8%

* Only patients with grade II ependymomas were included in the present study; 5 of the 75 patients screened were grade III ependymomas, see Methods. PE: pulmonary embolism, UTI: urinary tract infection.

**Table 3 cancers-15-03674-t003:** Detailed history of IME-II progressions.

Sex-Age	Spinal Segments	Preop. MCG	Extent of Resection	Ki-67	Delay of Progression	Symptoms	Delay of Retreatment	Management of Progression	Last FU (Months)	Last FU MCG
F-37	C7-T3	I	STR	8%	12	No	40	Surgery (STR)	86	I
H-59	T8-T9	I	STR	1%	48	No	NA	Conservative	67	I
F-68	C3-C6	II	STR	1%	18	Yes	22	Surgery (PR) + radiotherapy	51	IV
F-48	C1-C2	I	STR	1%	66	No	68	Radiotherapy	107	I
F-48	T2-T4	I	PR	3%	16	Yes	18	Surgery (GTR)	151	I
F-33	C4	I	STR	1%	81	No	NA	Conservative	103	I
F-39	C2-C4	I	STR	1%	35	No	NA	Conservative	186	I

## Data Availability

Anonymized clinical and radiological data are available upon request to the corresponding author.

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
