# Peer review of "Long-Term Outcomes after Incomplete Resection of Intramedullary Grade II Ependymomas: Is Adjuvant Radiotherapy Justified?"

_cancers, 2023, doi:10.3390/cancers15143674_

Round 1
Reviewer 1 Report
It is a very well written paper reporting on long-term follow up after the resection of intramedullary grade 2 ependymomas. The authors aim to answer a very controversially discussed and highly relevant issue: the need for adjuvant radiation therapy after incomplete tumor resection. By means of a restrospective analysis of 46 patients they conclude that the radiation therapy appears superfluous.
The number of the included patients seems to be low, however we are speaking about a relatively rare disease and the amount of the patients is adequate for a monocenter study. However, the number is not adequate to achieve any statistically relevant power to recognize any clear benefits or harms of the adjuvant treatment. In face of very slow growing tumor entity and its rare occurrence, a randomized prospective study is certainly not feasible. But at least a multicenter approach could be thinkable to get more validity. In any case, the results of the study do not allow the statements in the „Conclusions“, nor the proposed treatment algorithms. The suggestion to observe asymptomatic recurrences is particularly surprising: Wait&see is a justified procedure as long as the tumor remains stable. The term „recurrence“ means growing tumor and should be usually treated as commonly accepted. The authors should revise their conclusion without exaggeration of the impact of their results
Author Response
We agree that the number of patients is too low to provide sufficient statistical power to evaluate the benefit or not of an adjuvant radiotherapy. A multicenter study would certainly increase the number of patients but also the heterogeneity in terms of surgical technique and management. We included a paragraph discussing the design of the study and its limitation (lines 206-2013). We also mitigated our conclusions in the simple summary, abstract and conclusion paragraph (lines 229-243), and modified the Figure 3 accordingly. We added a paragraph discussing the rationale regarding the toxicity of RT to further support (lines 226-230) our strategy.
Reviewer 2 Report
The Authors have reported a single institution experience regarding the role of the observation in patients who underwent surgery for spinal ependymomas.
There are some remarks from my point of view:
Page 3 line 106. As spinal ependymoma patients that were included in this retrospective cohort range in a period from 2009 to 2018, which WHO Classification has been used for the neuropathological diagnosis? The 2016 or the 2021? In case of the Authors have used the WHO 2021, it is crucial to detail whether there are some patients with NF2 mutation or MYCN amplification, that have been suggested to have an impact on prognosis of spinal ependymomas.
Page 7 line 209. I disagree with the following sentence “Surgery is proposed only for symptomatic patients and…”. An intramedullary lesion should need to be histologically defined, as there are many other histological options instead of ependymomas. According to the EANO Guidelines (Rudà R, Reifenberger G, Frappaz D, Pfister SM, Laprie A, Santarius T, Roth P, Tonn JC, Soffietti R, Weller M, Moyal EC. EANO guidelines for the diagnosis and treatment of ependymal tumors. Neuro Oncol. 2018 Mar 27;20(4):445-456. doi: 10.1093/neuonc/nox166. PMID: 29194500; PMCID: PMC5909649) “Resection is recommended to obtain a histological diagnosis and should be a gross total resection whenever feasible. As the morbidity can be significant, detailed informed preoperative counseling by a surgeon experienced in performing such surgery is important”, class of evidence II, level of recommendation B. Please amend this sentence accordingly.
Overall, I suggest providing a more nuanced conclusion. Although I agree with the Authors that ependymoma recurrence often is asymptomatic and the preservation of the neurological functions and quality of life are extremely important, only a randomized study could address the question whether in case of either gross-total or subtotal resection the observation is not inferior over adjuvant radiotherapy. A single successful experience of one institution can generate a hypothesis that need to be proved in an adequate setting of a clinical trial. Moreover, the molecular biology should be included in this discussion for the reason that I have already aforementioned.
Moderate editing of English language required
Author Response
Page 3 line 106. As spinal ependymoma patients that were included in this retrospective cohort range in a period from 2009 to 2018, which WHO Classification has been used for the neuropathological diagnosis? The 2016 or the 2021? In case of the Authors have used the WHO 2021, it is crucial to detail whether there are some patients with NF2 mutation or MYCN amplification, that have been suggested to have an impact on prognosis of spinal ependymomas.
The 2016 WHO classification was used for neuropathological diagnosis. No molecular studies were performed routinely to identify specific mutations; hence we did not discuss this point. We added this point to the Methods (line 67)
Page 7 line 209. I disagree with the following sentence “Surgery is proposed only for symptomatic patients and…”. An intramedullary lesion should need to be histologically defined, as there are many other histological options instead of ependymomas. According to the EANO Guidelines (Rudà R, Reifenberger G, Frappaz D, Pfister SM, Laprie A, Santarius T, Roth P, Tonn JC, Soffietti R, Weller M, Moyal EC. EANO guidelines for the diagnosis and treatment of ependymal tumors. Neuro Oncol. 2018 Mar 27;20(4):445-456. doi: 10.1093/neuonc/nox166. PMID: 29194500; PMCID: PMC5909649) “Resection is recommended to obtain a histological diagnosis and should be a gross total resection whenever feasible. As the morbidity can be significant, detailed informed preoperative counseling by a surgeon experienced in performing such surgery is important”, class of evidence II, level of recommendation B. Please amend this sentence accordingly.
We modified this sentence to state that " Surgery is indicated for evolutive and/or symptomatic tumors and GTR should always be the aim of this initial surgery [EANO reference]". In our opinion, surgery should not be offered to asymptomatic patients unless there is proven radiological and/or clinical progression. For a newly diagnosed and asymptomatic tumor, we would propose an initial follow-up period given the neurological risks of surgery. We also added a reference to the EANO guidelines in the introduction (lines 54-59).
Overall, I suggest providing a more nuanced conclusion. Although I agree with the Authors that ependymoma recurrence often is asymptomatic and the preservation of the neurological functions and quality of life are extremely important, only a randomized study could address the question whether in case of either gross-total or subtotal resection the observation is not inferior over adjuvant radiotherapy. A single successful experience of one institution can generate a hypothesis that need to be proved in an adequate setting of a clinical trial. Moreover, the molecular biology should be included in this discussion for the reason that I have already aforementioned.
We agree that the number of patients is too low to provide sufficient statistical power to evaluate the benefit or not of an adjuvant radiotherapy. A multicenter study would certainly increase the number of patients but also the heterogeneity in terms of surgical technique and management. We included a paragraph discussing the design of the study and its limitation (lines 206-2013). We also mitigated our conclusions in the simple summary, abstract and conclusion paragraph (lines 229-243), and modified the Figure 3 accordingly.
Round 2
Reviewer 1 Report
The authors provided substantial revision of the manuscript which significantly increased the overall quality. I would still recommend some changes:
- please include clear recommendation of RT after partial resection, i.e <90% of the tumor to the flow chart in the discussion
- what does „if cleavage plan is unclear“ mean? Please, explain
- please discuss on potential value of the molecular genetics of the tumor independent of the WHO grade for the decision on adjuvant treatment (e.g. https://pubmed.ncbi.nlm.nih.gov/30053291/
Author Response
The authors provided substantial revision of the manuscript which significantly increased the overall quality. I would still recommend some changes:
- please include clear recommendation of RT after partial resection, i.e <90% of the tumor to the flow chart in the discussion
We added the following sentence in the discussion "Partial resections (< 90%) for IME-II are rare (13% in our series) and are often the result of an unusual presentation of the tumor limiting surgical resection. Such cases are therefore best treated with adjuvant radiotherapy." (line 203-206). Given the rare occurrence of such cases, we preferred not to include it in the flow chart since the discussion is mostly about subtotal resections.
- what does „if cleavage plan is unclear“ mean? Please, explain
We explained in the discussion: " Indeed, although IME-II are usually well circumscribed and amenable to gross total re-section, they are not encapsulated lesions and may sometimes be infiltrative [4,8,17]. In such cases, subtotal resection can be performed to maximize safety and will, in most cases, consist in leaving a thin tumoral layer wherever the cleavage plan is unclear." (lines 197-200). To avoid any confusion, we removed the note "if cleavage plan unclear" from Figure 3.
- please discuss on potential value of the molecular genetics of the tumor independent of the WHO grade for the decision on adjuvant treatment (e.g. https://pubmed.ncbi.nlm.nih.gov/30053291/)
We added the following paragraph to further discuss molecular studies in the management of ependymomas: "Recently, molecular studies of ependymomas has led to reappraisal of these tumors in the WHO 2021 classification [21]. In particular, MYCN amplification now defines a distinctive subtype of aggressive spinal ependymomas with early leptomeningeal dissemination and anaplastic morphology [22]. Moreover, spinal myxopapillary ependymomas are now classified as WHO grade II tumors since their prognosis is more consistent with a grade II behavior [21]. Lastly, methylation-based classification also allows reclassification of spinal ependymomas to molecular subgroups, although no significant clinical and prognostic features have yet emerged [23]. No doubt that future molecular studies will advance our understanding of spinal ependymomas and might refine our therapeutic strategy, notably regarding the indication of adjuvant radiotherapy." [lines 248-257]
Reviewer 2 Report
Thanks for providing point-by-point reply to queries of the reviewer.
Author Response
Thank you for the helpful comments